# Characterization of Roasting Time on Sensory Quality, Color, Taste, and Nonvolatile Compounds of Yuan An Yellow Tea

**DOI:** 10.3390/molecules27134119

**Published:** 2022-06-27

**Authors:** Fei Ye, Xiaoyan Qiao, Anhui Gui, Panpan Liu, Shengpeng Wang, Xueping Wang, Jin Teng, Lin Zheng, Lin Feng, Hanshan Han, Binghua Zhang, Xun Chen, Zhiming Gao, Shiwei Gao, Pengcheng Zheng

**Affiliations:** 1Institute of Fruit and Tea, Hubei Academy of Agricultural Sciences, Wuhan 430068, China; yf421@163.com (F.Y.); guianhui@tricaas.com (A.G.); liuppitea@163.com (P.L.); wwsspp0426@163.com (S.W.); wangxueping79-79@163.com (X.W.); jobbase@163.com (J.T.); caozi20121117@163.com (L.Z.); zuotianyujintian@163.com (L.F.); ttppuu@163.com (X.C.); 2Guangdong Provincial Key Laboratory of Tea Plant Resources Innovation and Utilization, Tea Research Institute, Guangdong Academy of Agricultural Sciences, Guangzhou 510610, China; qiaoxiaoyan@tea.gdaas.cn; 3MuLanTia Xiang Co., Ltd., Huangpi District, Wuhan 432200, China; it386574909@163.com; 4Danding Tea Company Limited, Danjiangkou Conty, Shiyan 442717, China; dandingchaye@163.com; 5Yuan’an Lei Zu Tea Company Limited, Yuan’an Conty, Yichang 444205, China; leizuchaye@163.com

**Keywords:** Yuan An yellow tea, roasting time, taste flavor, color, aroma, non-targeted metabolomics

## Abstract

Roasting is crucial for producing Yuan An yellow tea (YAYT) as it substantially affects sensory quality. However, the effect of roasting time on YAYT flavor quality is not clear. To investigate the effect of roasting time on the sensory qualities, chemical components, odor profiles, and metabolic profile of YAYTs produced with 13 min roasting, 16 min roasting, 19 min roasting, 22 min roasting, and 25 min roasting were determined. The YAYTs roasted for 22 min got higher sensory scores and better chemical qualities, such as the content of gallocatechin (GC), gallocatechin gallate (GCG), free amino acids, solutable sugar, meanwhile the lightness decreased, the hue of tea brew color (b) increased, which meant the tea brew got darker and yellower. YAYTs roasted for 22 min also increased the contents of key odorants, such as benzaldehyde, nonanal, β-cyclocitral, linalool, nerol, α-cedrol, β-ionone, limonene, 2-methylfuran, indole, and longiborneol. Moreover, non-targeted metabolomics identified up to 14 differentially expressed metabolites through pair-wise comparisons, such as flavonoids, phenolic acids, sucrose, and critical metabolites, which were the main components corresponding to YAYT roasted for 22 min. In summary, the current results provide scientific guidance for the production of high quality YAYT.

## 1. Introduction

Tea (*Camellia sinensis* L.) is one of the most traditional beverages consumed worldwide due to its health benefits, satisfactory taste, and aroma [1]. Tea can be classified into green, black, oolong, dark, white, and yellow tea based on different manufacturing processes. Yellow tea is one of the six major teas in China, and it is the second tea discovered after green tea. Its history can be dated back to the mid-Tang Dynasty between 618–907 Common Era, mainly produced in Hubei, Hunan, Anhui, Sichuan, Zhejiang, and Guangdong provinces. Recent studies have shown that yellow tea is one of the main sources of antioxidants [2] and has many other functions [3,4,5,6,7].

Yellow tea can be classified into the following categories according to the maturity of the tea leaves: large-leaf yellow tea, small-leaf yellow tea, and yellow bud tea [8]. Yuan An yellow tea (YAYT), one of the top Chinese small-leaf yellow tea, has a unique, tasty, caramel flavor, described as similar to crispy rice crust-like aroma. As we know, the unique flavor of YAYT is attributed to its specific processing technology. The traditional processing technology of yellow tea is spreading, fixing, rolling, yellowing, primary drying, and full roasting. Among these procedures, roasting plays an important role in fragrance formation and quality fixation of high-quality YAYT.

As far as yellow tea processing is concerned, researchers have been focusing on the improvement in the sensory and physico-chemical quality of yellow tea [2,9]. Roasting is an important manufacturing procedure and involves the development of sensory quality. There has been much roasting research involved in large-leaf yellow tea [9,10,11,12]; for instance, the old fire roasting was essential for the formation of large-leaf yellow tea flavor with strong roasted, nutty, woody odors and weak fatty, fruity odors, and retaining high levels of gallocatechin gallate (GCG), total volatiles and heterocyclic compounds [10]. However, until now, few studies have been concerned with the effect of roasting time on sensory quality, color, taste, and nonvolatile compounds of YAYT.

Until now, there are several roasting methods for natural product processing, for instance, hot air roasting, hot roller roasting, far-infrared drying, microwave drying, and so on. The above roasting methods have both advantages and disadvantages [13,14]. In the tea processing field, hot roller roasting methods have been in large-scale applications [15,16]. In fact, some new roasting methods, such as far-infrared drying and microwave drying, have not been in application for the quality improvement of yellow tea. Therefore, this study aimed to conduct a sensory evaluation and analyze the dynamic changes in chemical qualities and aroma components of YAYT roasted by hot roller method at different times, which could provide production guidance to manufacture stable and high-quality YAYT.

## 2. Materials and Methods

### 2.1. Preparation of Tea Samples

Tea samples were processed by an experienced tea worker in Leizu Tea Co., Ltd. (Yuan’an County, Hubei, China). The fresh tea leaves (cultivar ‘population’) were plucked in the Lu’Yuan area (Yuan An County, Hubei, China) on 3 April 2021. Tea samples were produced by an experienced tea master through the conventional manufacturing processes.

### 2.2. Methods of YAYT Processing Procedures

The detailed processing method: Fresh tea leaves were plucked and spread 6 h at a thickness of 2 cm until 100 g of leaves contained 70 g of water. The spread leaves were subjected to fixation for 2.5 min in a hot roller machine (6CST-80, Zhejiang Shang Yang Co., Ltd., Quzhou, China). The fixed leaves were subjected to rolling for 30 min in a roller machine (6CR-30, Zhejiang Shang Yang Co. Ltd., Quzhou, China). Subsequently, the rolled leaves were yellowed in an artificial climate box (RXZ-328A, Changzhou City Solid Germany Instrument Co., Ltd., Changzhou, China) at 30 °C for 6 h with airflow. The primary drying was conducted by a double-pan roasted machine at 150 °C for 20 min. Finally, the primary drying leaves were roasted under 130 °C by a newly designed roller machine with five different time treatments, including 13 min, 16 min, 19 min, 22 min, and 25 min, respectively. Five replicates were used for each time treatment sample.

### 2.3. Sensory Evaluation

The sensory tea quality was assessed by five professional tea tasters from the Institute of Fruit and Tea, Hubei Academy of Agricultural Sciences, China. The description and scores of the tea samples were evaluated according to the national standards for the methodology of the sensory evaluation of tea (GB/T 23376-2018) and tea vocabulary for sensory evaluation (GB/T14487-2017). Briefly, tea samples (3 g) were extracted with 150 mL of freshly boiled distilled water for 5 min. Tea infusions were individually presented in white porcelain bowls, and tea samples were blind-coded with random numbers. Then, panelists were instructed to smell and drink the tea infusions and pause for 30 s between the different samples [17,18]. Each sample was assessed three times via blind evaluation. The evaluation was completed by well-trained panelists (two males and three females aged 28−45 years). The quality scores of the tea samples were determined by calculating the averages of the scores from the five panelists. Each evaluation was randomly replicated five times on different days.

### 2.4. Determination of YAYT Color Quality

According to the concept of the *L, a, b* in three-dimensional color space, the *L* represents the lightness, with 100 for white and 0 for black. The *a* shows the redness when positive and the greenness when negative. Similarly, the *b* shows yellowness when positive and blueness when negative. The color of the tea, brew, and infused tea were determined by the hue determination using a spectrophotometer (CM-5, Konica Minolta (China) Investment Ltd., Shanghai, China) [9,14,15,19,20], and distilled water was used as a blank control.

### 2.5. Determination of YAYT Physico-Chemical Quality

Moisture content was analyzed by using the 120 °C drying methods (GB/T 8304-2013). The content of amino acids was analyzed by using the ninhydrin colorimetric method (GB/T5009. 124-2003). The content of total tea polyphenols was analyzed by using the iron tartrate colorimetric method (ISO14502-1:2005). Soluble sugar content was analyzed using the anthrone colorimetric method [21].

### 2.6. Analysis of YAYT Volatiles

The tea samples were powdered in an ultra-mill (A 11 basic analytical mill, Germany) using a 0.35 mm sieve. A solid-phase micro-extraction (SPME) fiber was exposed for 10 min in the injection port of the gas chromatography (GC) instrument at 280 °C to remove any remaining volatiles from the fiber before each extraction. A total of 3.0 g of the dry tea sample were added to a 100 mL vial sealed with silicone septa and infused with 30 mL of boiling water; afterward, 20 μL of internal standard solution (ethyl caprate) was added immediately. The vial was kept in a water bath at 50 °C for 10 min to equilibrate, and the SPME fiber was exposed for 50 min to the headspace while the sample was maintained at 50 °C. The fiber was then placed in the GC injector port and thermally desorbed at 240 °C for 3 min.

GC-MS analysis was performed on the Agilent 7890A GC interfaced with the Agilent 5975C MSD ion Trap MS. A DB-5MS capillary column (30 m × 0.25 mm × 0.32 µm) was used for separating. The GC oven temperature condition: 50 °C (held for 5 min) initially, increased to 180 °C (held for 2 min) at a rate of 3 °C/min, and finally increased to 250 °C (held for 3 min) at a rate of 10 °C/min. Helium (percentage purity > 99.999%) was used as a carrier gas at a constant flow rate of 1.0 mL/min. The MS was operated in the EI mode at an electronic energy of 70 eV. The injector and ion source temperature was 240 °C and 230 °C, respectively, and the MS was scanned at a range of 35–400 AMU. Compounds were tentatively identified using the National Institute of Standards and Technology (NIST) library (14.L). Also, the linear retention indices (RI) were calculated using n-paraffins C7–C40 as external references as described previously [7,10,14,22,23,24]. The concentration of the volatiles was calculated in µg/L based on an internal standard solution.

### 2.7. Analysis of the Composition of YAYT through Non-Targeted Metabolomics

To get an overview of the metabolic profile and also to confirm the results of sensory quality, physico-chemical quality, and volatile compounds of YAYT processed by the different roasted time treatments, non-targeted metabolomics was carried out as following steps. The experiments were performed on the ultra-high performance gas chromatography system (Agilent 6890A/5973C GC-MS). About 50 mg of the dry tea sample’s power was applied to the extraction procedure and extracted with 800 μL of methanol, then 10 μL of internal standard (2.8 mg/mL, DL-o- Chlorophenylalanine) was added. Chromatographic separation of the tea metabolome was carried out on the column (Agilent J&W Scientific, DB-5ms, 30 m × 0.25 mm × 0.25 μm). The injection condition: injector temperature 280 °C, ion source temperature 230 °C, quadrupole rod temperature 150 °C, Helium (high purity) >99.999%, injection mode splitless, injection volume 1.0 μL. The column temperature condition: was held at 70 °C (held for 2 min), increased to 200 °C at a rate of 10 °C/min, then increased to 280 °C (held for 6 min) at a rate of 5 °C/min. The column effluent was fully scanned in the mass at a range of 50–550 *m*/*z*.

The data was performed feature extraction and preprocessed with XCMS in R software (version 4.2), and then normalized and edited into a two-dimensional data matrix by excel software (version 2010), including retention time (RT), the mass-to-charge ratio (MZ), observations (samples), and peak intensity. Nine hundred sixty-six features were collected in this experiment, the data after editing was performed Multivariate Analysis (MVA) using SIMCA-P software (version 13.0, Umetrics AB, Umea, Sweden).

### 2.8. Statistical Analysis

The results were expressed as a mean of three measurements for the analytical determination. The analysis of significant differences between means was determined by one-way ANOVA (Duncan’s multiple range tests) using SPSS 23.0 (Demo version, Armonk, NY, USA). Figures were made by Origin 8.0 software (Demo version, Northampton, MA, USA). Principal component analysis (PCA), and orthogonal partial least-squares discrimination analysis (OPLS-DA) were conducted by SIMCA-P 13.0 software (version 13.0, Umetrics, Umea, Sweden). HCA was generated by the Multi Experiment Viewer (MEV) 4.9.0 (Oracle Corporation, Redwood Shores, CA, USA).

## 3. Results

### 3.1. Effect of Different Roasting Times on YAYT Sensory Quality

The sensory evaluation results of YAYT processed by five different roasting time treatments are listed in Table 1. The roasting time caused a distinct impact on the dry tea streak, dry tea color, liquor color, aroma, and taste of the YAYT. The samples roasted for 19 min and 22 min showed a curly tight, and heavy, bright yellow color, fried rice-like odors, mellow and thick taste flavor with high scores in dry tea color, streak, aroma, and taste, respectively (*p* < 0.05), but little on infused leaves. The samples roasted for 13 min and 16 min also showed a curly tight, and heavy, refreshing aroma and chestnut-like aroma, respectively, but a yellowish-green color. The samples roasted for 25 min had a curly tight, and heavy aroma, dark yellow color, and a bitter taste. The sensory evaluation results indicated that the samples roasted for 13 min, 16 min, and 25 min could not meet the quality requirements of YAYT. Thus, we deduced that the samples roasted for 19 min and 22 min were responsible for the formation of appearance and flavor characteristics in YAYT.

### 3.2. Effect of Different Roasting Times on YAYT Color Quality

The color quality results of YAYT processed by different roasted times were listed in Table 2. The different roasted time treatments also caused a distinct impact on the lightness and the hue of tea (*a* and *b*). The color of dried YAYT became redder and redder with the extension of roasting time, while there were no differences in lightness (*L*) and the hue of *b* of different roasting time treatments. With increasing roasting times, the tea brew became darker and darker; the hue of *a* firstly decreased and then increased, while the hue of *b* continued to increase. The results from this YAYT color evaluation were quite similar to other results [13,19,25,26].

### 3.3. Effect of Different Roasting Times on YAYT Physico-Chemical Quality

The chemical compositions of YAYT treated with different roasting times are shown in Figure 1. Eight polyphenols, including catechin (C), epicatechin (EC), GC, epicatechin gallate (ECG), epigallocatechin (EGC), GCG, and epigallocatechin gallate (EGCG), were identified and quantified in YAYT samples. With the extension of roasting time, the content of C, EC, ECG, EGC, EGCG, and total catechins decreased significantly, while the content of GCG increased, and the GC was firstly increased and then decreased. The content of free amino acids firstly increased and then decreased after 22 min roasting. As a consequence, the YAYT roasted for 22 min has the highest content of free amino acids and sugars.

### 3.4. Effect of Different Roasting Times on YAYT Volatile Compounds

Different roasting times had a great effect on the relative contents of volatile components. Eighty-nine volatile components were detected and identified, including twelve kinds of aldehydes, five kinds of alcohols, two kinds of ketones, three kinds of alkenes, four kinds of esters, and five kinds of other substances, among which alcohols and aldehydes were the main compounds. A comparison analysis revealed that YAYT treated with different roasting times contained the same constituents but different relative contents. These compounds were further classified into six categories and presented in Table 3.

Upon further analysis, compared with the other time treatments, the YAYT roasted by 22 min increased contents of benzaldehyde, nonanal, β-cyclocitral, linalool, nerol, α-cedrol, β-ionone, limonene, 2-methylfuran, indole, N-ethylpyrrole, 3-ethyl-2,5-dimethylpyrazine, and longiborneol. The above substances had a positive correlation with the sweet, flowery, or crispy rice-like flavor of tea [27,28], 2-methylfuran, and 3-ethyl-2,5-dimethylpyrazine could have a positive contribution to the roasted flavor of YAYT because of its extremely low threshold, which was probably derived from the Maillard reaction. While the YAYT roasted for 13 min increased contents of hexanal, heptanal, 1-octen-3-ol, geraniol, geranylacetone, styrene, and methyl sulfide, which had a positive correlation with the green and grassy flavor [29,30,31,32].

### 3.5. Metabolic Profiles of YAYT at Different Roasting Times

Based on the results of sensory evaluation, color quality, chemical components, and volatile compounds, the YAYT treated with 22 min roasting was considered to have the best quality. Therefore, the YAYT treated with 13 min and 22 min roasting were chosen for non-targeted metabolomics analysis; the 01 sample was used as a referrence. The PCA scores plot (R^2^X = 0.742, Q^2^ = 0.491), PLS-DA scores plot (R^2^X = 0.68, R^2^Y = 0.989, Q^2^ = 0.859), OPLS-DA scores plot (R^2^X = 0.619 R^2^Y = 0.946, Q^2^ = 0.808) and permutation test showed high reproducibility, reliability of the metabolomics results, and clear distinction between the treatments with two roasting time (Figure 2). The result also showed an evolving pattern, which indicated that different roasting time treatments influenced the chemical profile of YAYT tea. One thousand thirty-seven features were collected in this experiment, and a total of 14 differential metabolites with significant differences were identified in 22 min treatment YAYT compared to 13 min treatment YAYT (Figure 3A). Among these metabolites, 3 were up-regulated, and 11 were down-regulated, based on the criteria of VIP > 1, *p* < 0.05, and match score ≥ 700 (Figure 3B).

The differential metabolites are shown in Table 4. Eight free amino acids showed extremely significant changes in the two roasting treatments. The levels of most free amino acids, including gallic acid, hexadecanoic acid, malic acid, threonic acid, shikimic acid, and 2-keto-l-gulonic acid were significantly decreased in YAYT roasted at 22 min compared to YAYT roasted for 13 min, while the content of octadecanoic acid and d-Glucuronic acid were increased. As for soluble sugar, the content of sucrose and d-fucose were significantly decreased in YAYT treated with 22 min.

To further explore the impact of differential metabolites in YAYT flavor and discover internal relations between metabolites and different roasting times, a functional kyoto encyclopedia of genes and genomes (KEGG) analysis was conducted. Fourteen differential metabolites were involved in nine perturbed KEGG pathways, such as ascorbate and aldarate metabolism, inositol phosphate metabolism, galactose metabolism, fatty acid biosynthesis, fatty acid elongation in mitochondria, phenylalanine, tyrosine, and tryptophan biosynthesis, starch and sucrose metabolism, and flavonoid biosynthesis. As we know, the above perturbed metabolic pathways were related to taste and aroma.

To better observe the changes of differential metabolites of YAYT roasted through two different time treatments, a lording plot (Figure 2B) analysis was conducted and reflected that octadecanoic acid, d-glucuronic acid were the main components of YAYT roasted for 13 min, while sucrose, gallic acid, hexadecanoic acid, malic acid, threonic acid, shikimic acid, myo-Inositol, 2-Keto-l-gulonic acid, phytol, epicatechin, and d-Fucose were the main components of YAYT roasted for 22 min. Therefore, Threonic acid, shikimic acid, and epicatechin were the main characteristic metabolites of YAYT. The results of metabonomics, physico-chemical quality, and volatile compounds analysis could be the reasons for the flavor formation of YAYT.

## 4. Discussion

YAYT is an important yellow tea widely consumed in China. There are different roasting methods in tea processing, among which hot roller is a widely used method. In this study, we aimed to investigate the effect of roasting time on the sensory qualities, chemical components, odor profiles, and metabolic profiles of YAYTs.

Sensory evaluation revealed that the roasting with 22 min could better retain the color of YAYT than other time treatments. This phenomenon has also been observed in other materials using the hot roller roasting method [31,32]. For example, tea roasted by a hot roller was darker than other drying methods [25,33,34]. The underlying mechanism of this change in brew color yellowing of YAYT could be due to the heating effect during roasting, which caused a non-enzymatic yellowed reaction depending on heating temperature and time. Therefore, the hot roller roasted YAYT got yellower color with the extension of roasting time. But the color difference (*∆E*) and the tea (+b) do not correlate with sensory results. After a certain time, the longer the roasting time was extended, the poorer sensory quality the YAYT got. The result of this sensory evaluation was quite similar to the previous report [25,26], which provides a rich and complex YAYT tea processing.

As known to all, tea has many functions such as anti-oxidation, hypolipidemic and anticarcinogenic activities due to its polyphenols and amino acids, which are the important chemical components in tea and the main components [34,35,36]. In this study, the YAYT roasted for 22 min showed a significantly higher water extract and amino acid content than that of other treatments. Free amino acids and sugar are the major flavor components accountable for the sweet and umami taste of tea infusion [29]. They are also important substrates for the generation of volatiles and non-volatiles in the Maillard reaction [10]. The difference in free amino acids and sugar concentrations in five samples may be because of different degrees of the Maillard reaction through different roasting times in the YAYT roasting process. The results suggested that proper prolongation of roasting time could promote the content of soluble sugar and free amino acids, which were positively correlated with sweet and mellow. In contrast, the bitter components of YAYT after a 22 min roasting time decreased, such as tea polyphenols, EGCG, ECG, total catechins, and caffeine, which were positively correlated with bitter taste and astringency [37]. Researchers found that higher roasting temperature and longer roasting time could lead to more losses of total phenolic contents [26]. Therefore, our finding was consistent with the previous study. Based on the above information, we concluded that proper prolongation of roasting time within 22 min is beneficial to improving the YAYAT taste, especially in the increase of sucrose and free amino acids.

Tea volatiles are important components of tea aroma, which have a great impact on the sensory quality of tea. The long time roasting increased volatiles contents that had a positive correlation with the crispy rice crust-like aroma. It was likely that hot air drying could increase the contents of glucoside alcohols, resulting in a decrease in alcohol content and might be a good accelerator for the formation of sweet and crispy rice flavors. Under the action of heating, these components undergo hydrolytic reactions and turn into a free state and give out a crispy rice-like aroma. Thus, the drying process contributes to the formation of Yuan An yellow tea aroma.

## 5. Conclusions

In conclusion, the present study demonstrated that the roasting time had an important effect on the flavor quality of YAYT, which could be evaluated by the physico-chemical quality, including catechins, caffeine, total sugars and free amino acids, and volatile compounds, as well as odor profiles, color qualities, and sensory evaluation. The compound composition contributed to the health benefit and flavor properties of the tea product. YAYT with a 22 min roasting treatment contained higher sensory scores and better chemical qualities, meanwhile, the lightness decreased, and the hue of tea brew color (b) increased, which means the tea brew was getting darker and yellower. YAYTs roasted for 22 min also increased the contents of key odorants. Moreover, Non-targeted metabolomics data confirmed that flavonoids, phenolic acids, and sucrose were the main components corresponding to the roasted YAYT sample. In summary, the results provide scientific guidance for the production of high-quality YAYT.

There is a limitation needed to be discussed here. Only the roasting time, but not even the temperature, was investigated in this study. In fact, we have picked two parameters; one was roasting temperature, and the other was roasting time. A study on the effect of roasting temperature on the quality formation of YAYT is under preparation in another paper. Some form of factorial screening design within a design of experiments framework will be performed, which will perhaps be suggested for future investigations.

## Figures and Tables

**Figure 1 molecules-27-04119-f001:**
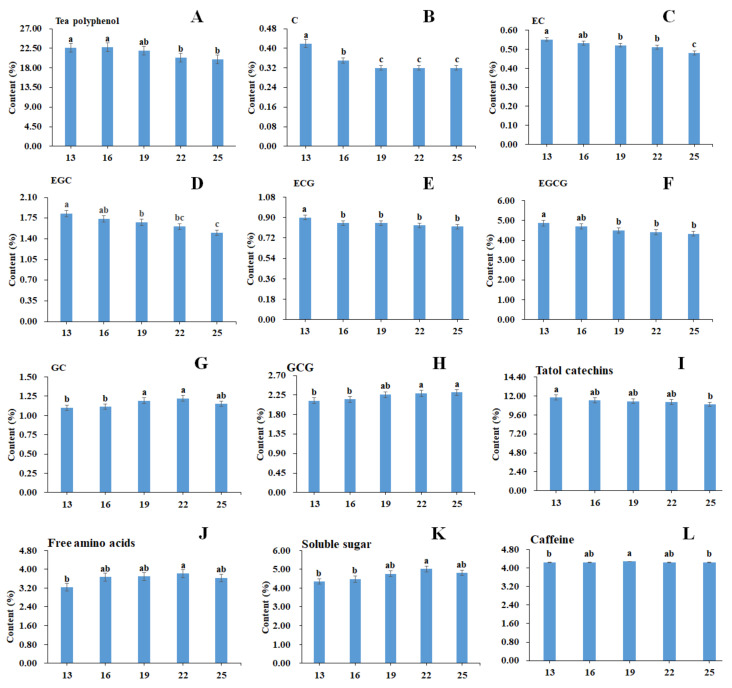
The content of tea polyphenol, catechins, total catechins, free amino acids, sugar, and caffeine in YAYT with different roasting times. (**A**) Tea polyphenol; (**B**) C; (**C**) EC; (**D**) EGC; (**E**) ECG; (**F**) EGCG; (**G**) GC; (**H**) GCG; (**I**) Total catechins; (**J**) Free amino acids; (**K**) Solutable sugar; (**L**) Caffeine. Columns labeled with ‘a’, ‘b’, and ‘c’ had significant differences (*p* < 0.05) from each other.

**Figure 2 molecules-27-04119-f002:**
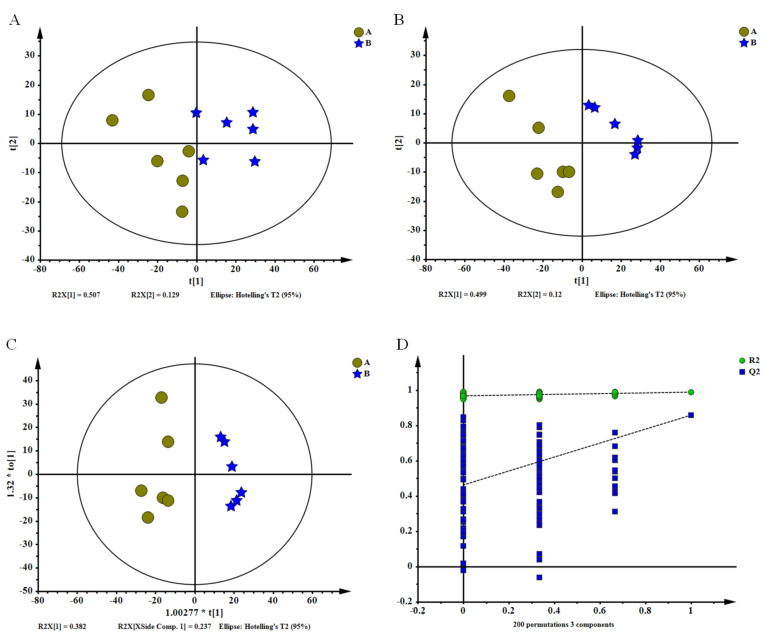
Group A, YAYT roasted for 13 min; Group B, YAYT roasted for 22 min. The vector value of R^2^ (0.748) and Q^2^ (0.668) from 200 permutations, which indicated that this model was not overfitting. Score plots from principal component analysis (**A**), partial least squares discrimination analysis (**B**), orthogonal partial least squares discrimination analysis (**C**), and its permutations test (**D**). The data set comprised 1037 filtered ion features. Spots in yellow indicate samples from 13 min treatment group, and stars in blue show samples from 22 min treatment group.

**Figure 3 molecules-27-04119-f003:**
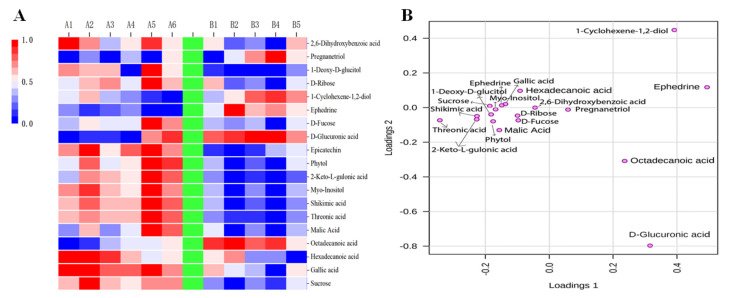
(**A**) Heat map of metabolites in YAYT treated with 13 min and 22 min roasting, respectively. Group A, YAYT roasted for 13 min; Group B, YAYT roasted for 22 min. Warm color and cold color indicated increased and decreased expression of the metabolites, respectively. (**B**) Loading plot of metabolites in YAYT treated with 13 min and 22 min roasting, respectively. Group A, YAYT roasted for 13 min; Group B, YAYT roasted for 22 min. Spots in pink indicate samples from group A, and spots in green show samples from group B, based on the criteria of VIP > 1, *p* < 0.05, and match score ≥ 700.

**Table 1 molecules-27-04119-t001:** Sensory quality scores of the YAYT with different roasting times.

Samples	Dry Tea Color (15%)	Dry Tea Streak (10%)	Liquor Color (10%)	Aroma (25%)	Taste (30%)	Infused Leaf (10%)	Total Scores
13 min	84.0 ± 0.5 ^c^	84.0 ± 0.6 ^c^	88.5 ± 0.8 ^c^	85.5 ± 0.5 ^b^	85.0 ± 1.0 ^b^	88.0 ± 0.5 ^a^	85.53 ± 0.5 ^c^
16 min	86.0 ± 1.2 ^b^	86.0 ± 0.5 ^b^	90.5 ± 1.0 ^a^	86.5 ± 1.2 ^b^	86.0 ± 1.1 ^b^	89.0 ± 1.0 ^a^	86.88 ± 0.7 ^b^
19 min	88.5 ± 0.9 ^a^	89.0 ± 1.0 ^a^	91.5 ± 0.8 ^a^	86.5 ± 0.7 ^a^	89.5 ± 1.3 ^a^	89.0 ± 1.0 ^a^	88.45 ± 0.4 ^a^
22 min	88.0 ± 0.6 ^a^	88.5 ± 1.2 ^a^	92.5 ± 1.1 ^a^	88.2 ± 0.8 ^a^	88.5 ± 1.2 ^a^	89.0 ± 1.0 ^a^	88.80 ± 0.9 ^a^
25 min	87.0 ± 1.1 ^b^	86.5 ± 0.6 ^b^	89.5 ± 1.1 ^b^	86.5 ± 1.0 ^b^	86.5 ± 1.0 ^b^	88.5 ± 0.5 ^a^	87.08 ± 0.5 ^b^

Note: the % values in the table mean percentages of total sensory scores. Data are presented as mean ± standard deviation (*n* = 5). Mean values with the different superscript letters (^a^, ^b^, ^c^) in the same column indicate a significant difference using a least significant difference (LSD) test (*p* < 0.05).

**Table 2 molecules-27-04119-t002:** Color quality scores of the YAYT with different roasting times.

Samples	Dry Tea Color				Brew Color			
	*L*	*a*	*b*	*∆E*	*L*	*a*	*b*	*∆E*
13 min	26.01 ± 1.56	−1.23 ± 0.095 ^a^	20.39 ± 1.76	33.07 ± 2.36	93.14 ± 0.056 ^a^	−4.64 ± 0.015 ^e^	15.13 ± 0.14 ^e^	94.48 ± 0.033
16 min	26.99 ± 1.18	−1.13 ± 0.095 ^b^	20.32 ± 1.38	33.80 ± 1.77	92.97 ± 0.076 ^b^	−4.81 ± 0.0095 ^b^	16.09 ± 0.13 ^b^	94.48 ± 0.053
19 min	26.93 ± 1.18	−1.12 ± 0.032 ^b^	20.74 ± 0.49	34.01 ± 0.45	92.96 ± 0.088 ^b^	−4.86 ± 0.0095 ^a^	16.03 ± 0.13 ^c^	94.42 ± 0.063
22 min	27.33 ± 0.56	−0.94 ± 0.050 ^c^	20.14 ± 0.55	33.97 ± 0.63	92.88 ± 0.049 ^c^	−4.77 ± 0.0095 ^c^	16.42 ± 0.45 ^a^	94.44 ± 0.076
23 min	26.42 ± 0.85	−0.74 ± 0.062 ^d^	19.89 ± 0.83	33.08 ± 1.18	92.84 ± 0.067 ^d^	−4.67 ± 0.0098 ^d^	16.45 ± 0.15 ^a^	94.43 ± 0.045

Data are presented as mean ± standard deviation (*n* = 5). Mean values with the different superscript letters (^a^, ^b^, ^c^, ^d^) in the same column indicate a significant difference using a least significant difference (LSD) test (*p* < 0.05).

**Table 3 molecules-27-04119-t003:** Main volatile compounds of the YAYT with different roasting times.

Category	Volatile Compounds	Threshold (μg/L) ^ψ^	Odor Description ^#^	Content/(μg·L)				
				13 min	16 min	19 min	22 min	25 min
Aldehydes	Hexanal	4.5	Green, grassy	1.53	1.35	1.29	1.17	1.18
	Heptanal	3	grassy, fresh	1.41	0.91	0.78	0.76	0. 74
	Benzaldehyde	350–3500	Almond-like, fruity, cherry-like, powdery, nutty	0.14	0.15	0.17	0.19	0.21
	Nonanal	1	Floral, Rose-like	1.23	1.58	1.45	1.35	1.07
	β-cyclocitral	32	lemon-like	0.44	0.77	0.67	0.58	0.42
Alcohols	1-octen-3-ol	10	Mushroom, sweet floral	0.39	0.32	0.21	0.19	0.13
	Linalool	6	Floral, sweet, grape-like, woody	0.46	0.62	0.56	0.52	0.28
	Nerol	300	Rose-like	0.71	0.75	0.75	0.81	0.65
	Geraniol	40–75	Rose-like, sweet, honey-like	1.02	0.93	0.96	0.72	0.45
	α-cedrol	-	woody	0.51	0.17	0.67	0.57	0.60
Ketones	Geranylacetone	60	Apple-like, fruity aroma	0.35	0.43	0.44	0.30	0.28
	β-ionone	0.007	Violet-like, raspberry, floral	0.54	0.70	0.79	0.56	0.40
Alkenes	Styrene	0.32	sweet	0.84	0.84	0.76	0.76	0.69
	limonene	10	Citrus, lemon, orange-like	0.21	0.30	0.35	0.45	0.20
Others	Methyl sulfide	0.3–1.0	Clean and refresh,	0.69	0.60	0.57	0.54	0.40
	2-methylfuran	0.001–0.004	Fired aroma	0.055	0.052	0.061	0.065	0.075
	N-ethylpyrrole	-	Roasted, caramel	0.028	0.032	0.038	0.045	0.056
	3-ethyl-2,5-dimethylpyrazine	0.0086	crispy rice-like	0.050	0.055	0.056	0.072	0.074
	Indole	140	Floral, animal-like	0.052	0.080	0.10	0.12	0.15
	longiborneol	-	woody	0.11	0.16	0.21	0.29	0.20

^#^ Odor description found in the literature with database (Flavornet; The LRI and Odour Database). ^ψ^ All the odor thresholds were obtained from: ’Odour & Flavour Detection Thresholds in Water (In Parts per Billion, μg/L)’.

**Table 4 molecules-27-04119-t004:** The differential metabolites of YAYT with different roasting times.

NO.	RT/min	Name	Match	VIP	*P*/T-Test	log_2_FC (13 min/22 min)
1	26.64	Sucrose	953	1.43	0.0003	0.2184
2	17.12	Gallic acid	938	1.40	0.0007	0.1577
3	18.46	Hexadecanoic acid	938	1.09	0.0209	0.1385
4	21.12	Octadecanoic acid	915	1.25	0.0049	−0.2977
5	11.71	Malic acid	913	1.05	0.0279	0.1552
6	12.52	Threonic acid	913	1.51	0.0001	0.3970
7	15.36	Shikimic acid	913	1.49	0.0001	0.2585
8	18.79	Myo-Inositol	889	1.44	0.0003	0.1875
9	14.86	2-Keto-l-gulonic acid	876	1.46	0.0002	0.2642
10	20.08	Phytol	857	1.29	0.0030	0.1880
11	29.70	Epicatechin	816	1.49	0.0001	0.1877
12	22.93	d-Glucuronic acid	789	1.05	0.0281	−0.4420
13	35.34	d-Fucose	733	1.06	0.0267	0.0978
14	4.30	Ephedrine	705	1.47	0.0001	−0.5730

Note: Fold change (13 min/22 min) is the logarithm of the ratio of the mean value of YAYT treated with 13 min to that of YAYT roasted for 22 min (Logarithms based on 2). The positive sign indicates that YAYT roasted for 13 min increases relative to YAYT roasted for 22 min, and the negative sign indicates a decrease.

## Data Availability

Not available.

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
