# Peer review of "Characterization of Roasting Time on Sensory Quality, Color, Taste, and Nonvolatile Compounds of Yuan An Yellow Tea"

_molecules, 2022, doi:10.3390/molecules27134119_

Round 1
Reviewer 1 Report
The article presents results on the effect of roasting time on the quality of yellow tea. This is a contribution to know the optimal process-quality conditions for this product.
In this work, an analysis of several factors that affect the quality of yellow tea is carried out, which is undoubtedly a contribution, but an integrated analysis of the results should be included, for example, the color difference (∆E), does not correlate with sensory results, the same goes for sensory and instrumental color results (specifically (+b))
It would be interesting to include a proposal on the effect of roasting on the increase in sucrose and free amino acids.
The volatile compounds that are generated in the roasting process should be associated with the sensory perception of yellow tea, however, the increase in a group of volatile compounds does not always have an effect on the perception of a food.
The conclusion “The current results provide a scientific basis to understand the reactions that occur during the YAYT drying process” seems too ambitious, since an analysis by the authors of the reactions involved in the process is not carried out in the document.
Other observations
In the presentation of the results, the sample number should be modified by the roasting time.
The abbreviations used must be defined, for example catachín (C); epicatechin (EC)
Check if the format of the journal allows results and discussion separately.
The authors must carry out an integrated analysis of the results, for example.
-Correlate the analysis of the yellow color (+b) with the sensory results.
-Differences in aroma could be associated with the amount of volatile compounds and their perception threshold.
Author Response
Reviewer 1
- In this work, an analysis of several factors that affect the quality of yellow tea is carried out, which is undoubtedly a contribution, but an integrated analysis of the results should be included, for example, the color difference (∆E), does not correlate with sensory results, the same goes for sensory and instrumental color results (specifically (+b))
RE: Thank you very much for your nice suggestion. We have compared that the tea and brew color difference (∆E) of different roasting treatments, and there were no significant difference between them (LSD test p < 0.05), and the color difference (∆E) was not correlated with sensory results, the same as the hue of tea(+b). We have added this information in the revised manuscript.
- It would be interesting to include a proposal on the effect of roasting on the increase in sucrose and free amino acids.
RE: Thank you very much for your nice suggestion. We have added a proposal on the effect of roasting on the increase in sucrose and free amino acids within 22 minutes in the Disccusion section in the revised manuscript.
- The volatile compounds that are generated in the roasting process should be associated with the sensory perception of yellow tea, however, the increase in a group of volatile compounds does not always have an effect on the perception of a food.
RE: Thank you very much for your nice suggestion. Volatile compounds 2-methylfuran and 3-ethyl-2,5-dimethylpyrazine could have a positive contribution for roasted flavor in YAYT, because of its extremely low threshold, which was probably derived from the Maillard reaction. The longer the Maillard reaction time, the more these two aroma compounds are formed, therefore, our finding is consistent with the sensory quality results. We have added this information in the Result section.
- The conclusion “The current results provide a scientific basis to understand the reactions that occur during the YAYT drying process” seems too ambitious, since an analysis by the authors of the reactions involved in the process is not carried out in the document.
RE: Thank you very much for your nice suggestion. “The current results provide a scientific basis to understand the reactions that occur during the YAYT drying process” had been replaced by “In summary, the current results provide a scientific guidance for production high quality YAYT.
- In the presentation of the results, the sample number should be modified by the roasting time
RE: Thank you very much for your nice suggestion. We have modified sample number into the roasting time in the revised manuscript
- The abbreviations used must be defined, for example catachín (C); epicatechin (EC)
RE: Thank you very much for your nice suggestion. We have defined all the abbreviations in the revised manuscript.
- Check if the format of the journal allows results and discussion separately.
RE: Thank you very much for your nice suggestion. We have already checked it, Molecules do not have strict formatting requirements, but all manuscripts must contain the required sections: Author Information, Abstract, Keywords, Introduction, Materials & Methods, Results, Conclusions, Figures and Tables with Captions, Funding Information, Author Contributions, Conflict of Interest and other Ethics Statements.
- The authors must carry out an integrated analysis of the results, for example. Correlate the analysis of the yellow color (+b) with the sensory results. Differences in aroma could be associated with the amount of volatile compounds and their perception threshold.
RE: Thank you very much for your nice suggestion. We have carried out an integrated analysis of the results in the revised manuscript as your suggestion.
Reviewer 2 Report
attach file

Author Response
- p.2, line 91: … under 130℃ by a new... change by .... under 130 ℃ by a new.
RE: Thank you very much for your nice suggestion. We have changed this mistake in the revised manuscript.
- p.6, line 223 ... indole,N-ethylpyrrole... change by ... indole, N-ethylpyrrole,...
RE: Thank you very much for your nice suggestion. We have changed this mistake in the revised manuscript.
Reviewer 3 Report
The manuscript is very interesting, I have the following comments:
1- On what basis, do authors decide the experimental times of roasting? Why no test was done at higher temperatures than 22 min?
2- Are there any quality assurance parameters for the chemical analysis done in this study?
Author Response
Reviewer 3
- On what basis, do authors decide the experimental times of roasting? Why no test was done at higher temperatures than 22 min?
RE: Thank you very much for your nice suggestion. Before these formal experiments, we have already done the pre-experiment. In the pre-experiment, nine roasting time treatments were set up, including 7 min, 10 min, 13 min, 16 min, 19 min, 22 min, 25 min, 28 min, 31 min. After comparing the results of the preliminary experiment, we selected 13 min, 16 min, 19 min, 22 min, 25 min roasting time treatments in the end. We did test at higher temperatures than 22 min, this treatment made YAYT dark yellow color and bitter taste, which did not meet the quality of good tea.
- Are there any quality assurance parameters for the chemical analysis done in this study?
RE: Thank you very much for your nice suggestion. The chemicals were analyzed according to national standards, and reference materials used for analysis of the chemicals were purchased from Sigma. In addition, chemical quality test were done in the institute of fruit and tea, Hubei academy of agricultural Sciences, which have rich experiences in detecting physical and chemical quality of tea, and metabolomics test was performed by professional companies. Based on the above condition, quality assurance parameters for the chemical analysis are absolutely guaranteed.
Reviewer 4 Report
The authors provide interesting research into one step of yellow tea processing, namely the roasting. My major criticism would be that from the multivariate process of tea processing, one single parameter was picked and investigated. Only the roasting time, but not even the temperature, was investigated. Such an experimental approach might not detect important factors, such as the interactions between several parameters. If I would have been involved in planning such a research, I would have suggested some form of factorial screening design within a design of experiments (DOE) framework. This limitation must be more clearly pointed out in the discussion, and DOE perhaps suggested for future investigations.
The following revisions are necessary:
Line 21 and throughout: add space between number and unit
Line 23: explain abbreviations
Line 27: lower case Non-targeted
Line 29: “asv” should read “as”?
Line 27-31: this extremely long sentence is difficult to understand. Please revise.
Line 43: “is one of major source antioxidants”. Not understandable, revise
Line 44-45: delete claims about health effects. They are mostly not backed up by clinical or epidemiological data.
Line 54: “is concerned”
Lines 78-79: are these the step described in the next sections? The methods section is a bit unaligned.
Line 81: did you use the powdered samples for sensory assessment as well? This is not the typical practice?
Line 91: is there a rationale why the temperature was not varied as well?
Line 147: where does the feces comes from? This is not mentioned in the previous sections. Did the study contain a clinical part with probands drinking tea of which the feces was analyzed? This must be included, also state ethical clearance.
Line 182: avoid “speculations”. In this case, I would use “deduced”
Figure 1: the font size is unreadably small, the resolution is also very bad
Line 285: I would mention the information about the variety in the methods section.
Author Response
Reviewer 4
- The authors provide interesting research into one step of yellow tea processing, namely the roasting. My major criticism would be that from the multivariate process of tea processing, one single parameter was picked and investigated. Only the roasting time, but not even the temperature, was investigated. Such an experimental approach might not detect important factors, such as the interactions between several parameters. If I would have been involved in planning such a research, I would have suggested some form of factorial screening design within a design of experiments (DOE) framework. This limitation must be more clearly pointed out in the discussion, and DOE perhaps suggested for future investigations.
RE: Thank you very much for your nice suggestion. We have picked two parameters, one is roasting temperature, and the other is roasting time, the study of effect of roasting temperature on quality formation of YAYT is under preparation. In this study, we only explored influence of roasting time on YAYT quality, and the limitation was pointed out in the Discussion in the revised manuscript as your suggestion.
- Line 21 and throughout: add space between number and unit
RE: Thank you very much for your nice suggestion. We have added space between number and unit in the revised manuscript.
- Line 23: explain abbreviations
RE: Thanks for your comments. We have explained abbreviations in the revised manuscript
- Line 27: lower case Non-targeted
RE: Thanks for your suggestion. We have modified Non-targeted to non-targeted in the revised manuscript
- Line 29: “asv” should read “as”?
RE: Thank you very much for your nice suggestion. Yes, “asv” should read “as”. We have modified the error.
- Line 27-31: this extremely long sentence is difficult to understand. Please revise.
RE: Thank you for your nice suggestion. We have revised this sentence as “Moreover, non-targeted metabolomics identified up to 14 differentially expressed metabolites through pair-wise comparisons, such as flavonoids, phenolic acids, sucrose and critical metabolites, which were the main components corresponding to YAYT roasted with 22 min.” in the revised manuscript.
- Line 43: “is one of major source antioxidants”. Not understandable, revise
RE: Thank you for your nice suggestion. We have replaced this sentence with “Yellow tea is one of the main sources of antioxidants”
- Line 44-45: delete claims about health effects. They are mostly not backed up by clinical or epidemiological data.
RE: Thank you for your nice suggestion. We have deleted these sentences.
- Line 54: “is concerned”
RE: Thank you for your nice suggestion. We have changed “are concerned” into “is concerned”.
- Lines 78-79: are these the step described in the next sections? The methods section is a bit unaligned.
RE: Thank you for your nice suggestion. Yes, you are right. We have deleted these steps in this section in the revised manuscript.
- Line 81: did you use the powdered samples for sensory assessment as well? This is not the typical practice?
RE: Thank you for your nice suggestion. The tea samples were used for sensory tasting, while the tea samples powder were used for physical and chemical analysis according to national standard.
- Line 91: is there a rationale why the temperature was not varied as well?
RE: Thank you for your nice suggestion. We have already answered this question, the roasting temperature here is the optimal temperature for high quality YAYT. We have picked two parameters, one is roasting temperature, and the other is roasting time, the study of effect of roasting temperature on quality formation of YAYT is under preparation. In this study, we only explored influence of roasting time on YAYT quality.
- Line 147: where does the feces comes from? This is not mentioned in the previous sections. Did the study contain a clinical part with probands drinking tea of which the feces was analyzed? This must be included, also state ethical clearance.
RE: Thank you for your nice suggestion. The feces means the dry tea samples power, and the study do not contain a clinical part, so ethical clearance is not applicable, and we have replaced feces into dry tea samples power in the revised manuscript.
- Line 182: avoid “speculations”. In this case, I would use “deduced”
RE: Thank you for your nice suggestion. We have changed speculations into deduced as your suggestion.
- Figure 1: the font size is unreadably small, the resolution is also very bad
RE: Thank you for your nice suggestion. We have enlarged the font and improved resolution in Figure 1 in the revised manuscript.
- Line 285: I would mention the information about the variety in the methods section.
RE: Thank you for your nice suggestion. We have mentioned the information about the variety in the methods section.
Round 2
Reviewer 4 Report
All my previous comments were adequately considered
Author Response
-
This manuscript is a resubmission of an earlier submission. The following is a list of the peer review reports and author responses from that submission.